# Covariance predicts conserved protein residue interactions important for the emergence and continued evolution of SARS-CoV-2 as a human pathogen

**William P. Robins❶\*, John J. Mekalanos**

Department of Microbiology, Harvard Medical School, Boston, Massachusetts, United States of America

\* william_robins@hms.harvard.edu

**Data Availability Statement:** All relevant data are within the paper and its Supporting information files.

## Abstract

SARS-CoV-2 is one of three recognized coronaviruses (CoVs) that have caused epidemics or pandemics in the 21st century and that likely emerged from animal reservoirs. Differences in nucleotide and protein sequence composition within related β-coronaviruses are often used to better understand CoV evolution, host adaptation, and their emergence as human pathogens. Here we report the comprehensive analysis of amino acid residue changes that have occurred in lineage B β-coronaviruses that show covariance with each other. This analysis revealed patterns of covariance within conserved viral proteins that potentially define conserved interactions within and between core proteins encoded by SARS-CoV-2 related β-coronaviruses. We identified not only individual pairs but also networks of amino acid residues that exhibited statistically high frequencies of covariance with each other using an independent pair model followed by a tandem model approach. Using 149 different CoV genomes that vary in their relatedness, we identified networks of unique combinations of alleles that can be incrementally traced genome by genome within different phylogenic lineages. Remarkably, covariant residues and their respective regions most abundantly represented are implicated in the emergence of SARS-CoV-2 and are also enriched in dominant SARS-CoV-2 variants.

## Introduction

The prior emergence of SARS-CoV and MERS-CoV as human pathogens is attributed to zoonotic viruses that transferred from bats to civets and camels, respectively, while SARS-CoV-2 is most similar to viruses isolated from both bats and pangolins [1–6]. The ~30kb genome size of all SARS-related CoVs renders sequence alignment and pairwise distance methods effective for phylogenic studies and determining genetic events that correlate with their adaption to the human host. While nucleic acid sequence-based phylogenies are informative, they clearly have limitations as not all single nucleotide polymorphisms are equal. For example, single-strand RNA viruses possess significant base-pairing in regions of their genomes that can result in different fitness costs even for synonymous mutations because higher-ordered RNA structures

**Funding:** This research was financially supported by both the National Institutes of Health Grant AI-018045-JJM and a generous contribution provided by The Morningside Foundation (WPR and JJM). The sponsors did not play any role in the study design, data collection and analysis, decision to publish, or preparation of this manuscript. www.nih.giv https://morningside.com/.

**Competing interests:** The authors have declared that no competing interests exist

**Abbreviations:** AA, Amino acid; ACE-2, angiotensin-converting enzyme-2; ATDS, Average Taxonomic Distribution Score; CoV, Coronavirus; CTD, Carboxy-terminal domain; FCS, Furin Cleavage Site; FP, Fusion Peptide; GISAID, Global Initiative on Sharing All Influenza Data; hACE-2, human-angiotensin converting enzyme-2; NCBI, National Center for Biotechnology Information; nsp, Nonstructural protein; nt, nucleotide; NTD, Amino-terminal domain; PCA, Principal Component Analysis; RBD, Receptor binding domain; SNV, Single nucleotide variant; WHO, World Health Organization.

and non-coding regions can impact replication, transcription, and recognition by the host immune system [7, 8]. Nucleotide polymorphisms also distinctly influence encoded amino acid [AA] residues depending on their position in a codon and thus further protein expression is often correlated with codon frequency or cognate tRNA abundance [9]. Similarly, codon usage has been studied in the context of mutational pressure and natural selection [10–13]. For example, the presence of repeating rare codons in the SARS-CoV-2 spike protein corresponding to the furin cleavage site (FCS) has been explored as circumstantial evidence for genetic manipulation [14]. Mutational pressure rather than translational selection is however reported in other work to be one dominant factor in the observed codon usage in other human RNA viruses [15, 16]. In RNA viruses, the host immune system can impose additional selective pressure on genomic nucleotide content; thus the high frequency of A- and U-ending codons and underrepresentation of CpG dinucleotides in RNA viruses including CoVs is attributed primarily to cytosine deamination and the pressure exerted by innate immune mechanisms [17–19]. Comparative analysis of SARS-CoV-2 to closely related CoVs suggests that C-to-U conversion played a significant role in the evolution of the SARS-CoV-2 [20]. In sum, codon usage, nucleotide sequences, and the AA content may affect virus adaptation with the latter being most relevant to protein folding, stability, function, and adaptive immune recognition in the host.

Many previous and ongoing efforts to study human CoV virus-host interactions are focused on AA residues and domains within the quaternary and tertiary structure of the spike trimer protein. For SARS-CoV, the stepwise adaption from the ancestral bat CoV to variants that infect civets, humans, and even laboratory mice is well-understood and can be traced to the domains in the spike protein that confer specificity to the host, especially in the context of residues within the receptor-binding domain (RBD) [3, 21, 22]. An ancestor to SARS-CoV-2 is yet to be established with certainty, but both residues within the spike RBD that interact with the host receptor angiotensin-converting enzyme 2 (ACE-2), and a furin cleavage site are believed to have contributed to its adaption to the human host and its enhanced transmission [23–25]. RaTG13, one of the closest bat CoV relatives to SARS-CoV-2 that shares ~96% nucleotide identity [26], is measured to have a reduced affinity for human ACE-2 (hACE-2) when both are compared and this is in part conferred by residues in spike [27]. However, molecular evidence for ACE-2 affinity being the primary determinant in host specificity for CoVs is also confounding. SARS-CoV and SARS-CoV-2 viruses that infect human cell lines via the hACE-2 receptor are found to vary in their ability to infect bat cell lines suggesting that the host range of β-coronaviruses is not only specified by spike RBD-hACE-2 interactions [24, 28].

SARS-CoV-2 and its close relative RaTG13 and other related β-coronaviruses are remarkably similar in nucleotide and amino acid identity [6, 20, 29]. However, this has not yet revealed a verifiable source or clear zoonotic emergence pathway as was accomplished with SARS-CoV [30]. Detected probable recombination events between coronaviruses isolated from both bats and pangolins established the hypothesis for the emergence of a mosaic virus, possibly through an intermediate host [20, 31, 32]. In this model, the identification of the SARS-CoV-2 progenitor would require the identification of ancestral viruses, resolved recombination positions, and acquired secondary mutations that contributed to its emergence. Independent of recombination, the amino acid sequence dissimilarity between SARS-CoV-2 and the most closely related viruses is still largely a consequence of single nucleotide variations (SNVs). However, other differences may be more complex and even suspect such as the unique presence of the spike FCS sequence [25]. Though FCSs are already recognized in the spike proteins in other coronaviruses [33, 34], this is not yet identified in any lineage B β-coronaviruses outside of SARS-CoV-2. In summation, none of the available virus genomes identified in natural hosts and laboratory collections can yet be applied to completely recapitulate a progenitor

or single source with absolute confidence based on the sequence of the entire SARS-CoV-2 genome.

In contrast to looking backward towards an ancestor, we reason the interminable acquisition of mutations in key host range proteins such as spike during the ongoing pandemic indicates continued selection and adaptive evolution which is also informative. Evidence for the adaptability and the plasticity of spike protein domains has been documented by the existence of single and multiple mutations that have been enriched in newly dominant variant lineages during the ongoing pandemic. For example, the spike *D164G* allele, a stand-alone defining mutation of the dominant SARS-CoV-2 A2a clade that emerged early in the pandemic, has been demonstrated to increase viral fitness and infectivity, possibly by influencing proteolytic processing, incorporation of spike protein in the virion, and conformational states of the spike protein [35, 36]. Subsequent dominant emerging variants within this clade notably possess additional mutations in spike and other genes. Moreover, a broad and diverse collection of variant mutations are represented independently in other distinct lineages [37, 38]. In spike, some of these mutations are attributed to immunity evasion, host-receptor interactions, and spike structure and conformational dynamics [39]. Distinct attributes or roles of each of these mutations are yet to be entirely elucidated and the actual extent of the contribution of any allele may be intricate. Importantly, the summative contribution of each of these single mutations in the context of other mutations is not yet known, and in general, the effects of mutations in many proteins other than spike have been less studied or are completely unknown.

Here we report the results of an investigation that sought to use the evolutionary history of sarbecoviruses to identify the most-conserved interactions between AA residues in key proteins encoded by CoV viruses most related to SARS-CoV and SARS–CoV-2. Specifically, we identified all covariant AA pairs and also larger correlated tandem model-based networks (clusters) of AA residues that exhibited statistically high frequencies of covariance with each other. We examined conserved covariance between protein sequences to uncover new insights into CoV evolution through the identification of apparent inter and intra-protein interactions. We propose that these covariant interactions of residues are important for virus evolution and may drive adaption to other hosts and influence transmission and pathogenicity by emergent variant viruses.

## Results

### Estimated phylogeny of β-coronaviruses with completed genomes

We first compare the extent of the evolutionary relatedness of 169 β-coronaviruses using whole-genome nucleotide sequences as phylogeny is expected to be correlated with residue covariance (Fig 1A and 1B). We postulated covariant amino acid (AA) residues play diverse roles in viral protein structure, interactions, and functions or instead may be a consequence of mutational accumulation and drift that is not biologically relevant to viral protein function. Phylogeny and relatedness of genomes are recognized to bias observed apparent coupling of AA mutations and influence covariation [40–44]. By generating a phylogenic tree to assist in identifying such phylogenic effects, AA variability at covariant residues can also be traced using tree topology and even branch length and therefore analyzed in the context of evolution [44, 45].

We aligned the deposited nucleotide [nt] sequences of 169 unique lineage β-coronaviruses between the initiation codon of *Nsp1* of the 16 polyprotein-encoding gene *ORF1ab* through to the termination codon of the *N* gene that encodes the nucleocapsid protein (Genome Accession numbers listed in S1 Table). The 30,553 nt gapped alignment of these strains spans all core and accessory genes except for the hypothetical gene *ORF10* downstream of *N. ORF10* is

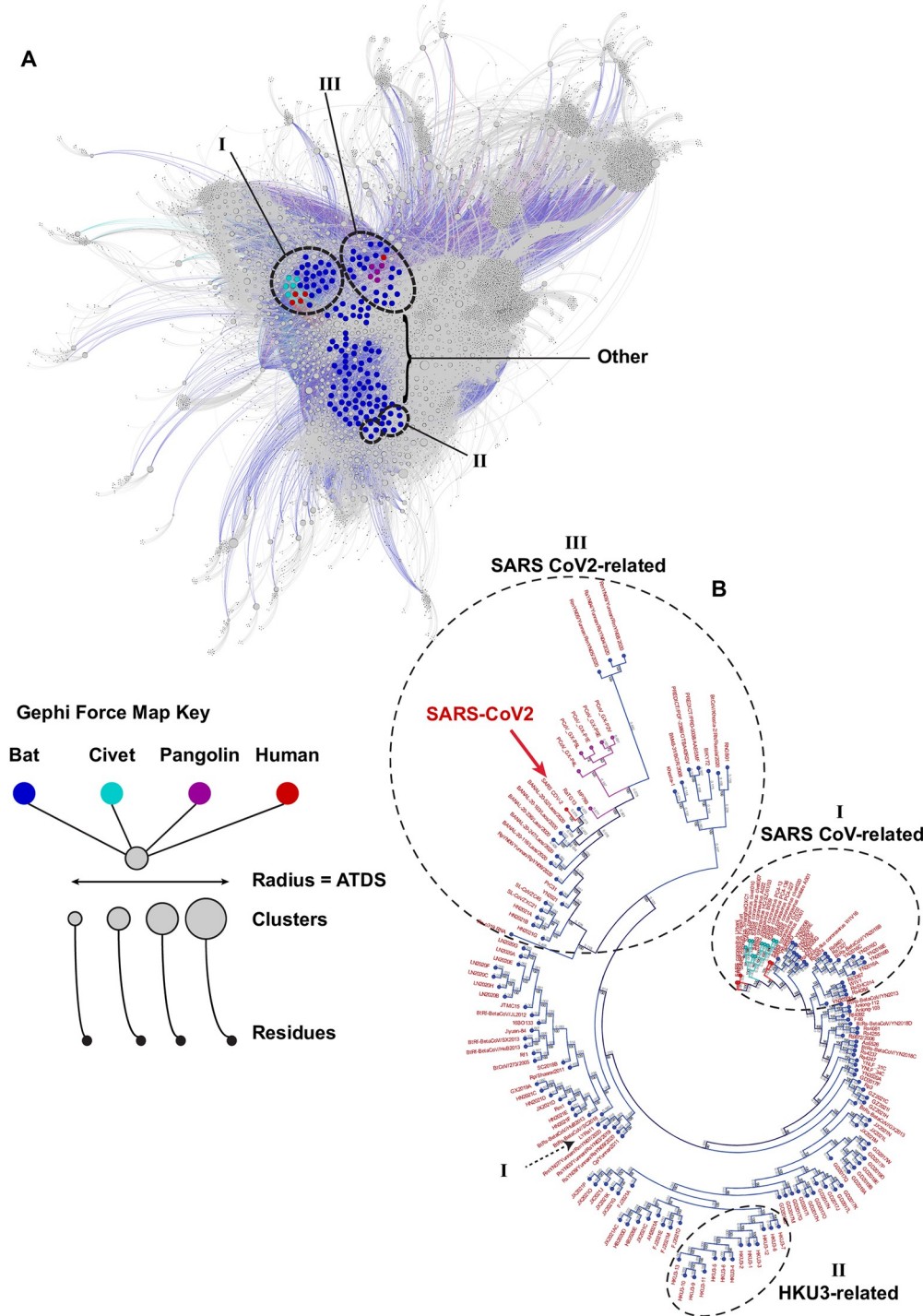

**Fig 1. Force mapping graph cluster based on amino acid residue covariance and nucleotide alignment-based ML tree phylogeny both reveal similar relationships.** (A) Gephi force mapping graph (Multiforce ForceAtlas 2) shows 149 CoVs based on all predicted clusters of covariant residues. The respective host for each CoV is indicated by color and the average taxonomic distribution score (ATDS) for each cluster is indicated by cluster circle size. CoVs most closely related to SARS-CoV, SARS-CoV-2, and HKU3 based on phylogeny are circled and labeled. (B) Overview of ML tree of 169 CoVs that spans nucleotides of genes *ORF1a/b* through *N* (Nucleocapsid). CoVs are colored by the host. SARS-CoV, SARS-CoV-2, and HKU3-related CoVs are circled to match groups in Fig 1A.

not predicted to be conserved in all represented CoVs in this group and is not essential for SARS-CoV-2 *in vitro* or *in vivo* [46]. In their entirety, CoVs represented in our analysis were isolated from bats (147), Civets (10), Pangolins (6), and humans (6). SARS-CoV is notably represented as both civet and human isolates and SARS-CoV-2 is represented by an initial reference sequence isolated in Wuhan during December 2019.

The topology and structure of the maximum likelihood (ML) tree generated by our alignment are largely consistent with published work using aligned variable regions of the genome [26, 31, 47–50]. Though certain genomic regions are predicted to be prone to recombination events that can lead to mosaicism during the evolution of CoVs [51–54], we did not alter our phylogenic analysis based on the exclusion or inclusion of any single core gene or gene region, apart from hypothetical gene *ORF10*. In an unrooted tree, distinct lineages are apparent; several CoVs are most related to SARS-CoV, a set of sublineages of CoVs more closely related to SARS-CoV-2, and a third distinct group of β-coronaviruses more closely related to HKU-3. Only a small subset of viruses in this analysis are predicted to be most closely related to SARS-CoV and SARS-CoV-2 relative to all other CoVs represented in this phylogeny. How these viruses differ from those more distantly related in the context of amino acid residue covariance was one aim of the comparative analysis presented here.

The relatedness of SARS-CoV-2 to certain bat and pangolin CoVs supports the emergence of this virus from a zoonotic reservoir. Using our nucleotide alignment-based tree, we identify RaTG13 and other more recently identified bat CoVs from Laos to be most closely related to SARS-CoV-2, designated as Group 1. Clusters of other pangolin and bat CoVs, some nearly clonal, comprise the next tier of related assemblages designated as Group 2 and 3, respectively. A small number of other CoVs, designed Groups 4 and 5, are less closely related to those in Groups 1–3 CoVs but also similarly distinct from other SARS-CoV and HKU-3-related viruses in our generated phylogeny. All CoVs in groups 1–5 are more likely to share a common ancestor with SARS-CoV-2 and we propose mutational and/or recombination events and also selective processes have generated the observed diversity within this subclade (Fig 2A and 2B).

## Alignment of conserved proteins in β -coronaviruses

We selected 149 CoVs represented in our phylogenic reconstruction based on the availability of annotated proteins and aligned the amino acids of core proteins using identified open reading framed (ORFs) common to all genomes. This includes the conserved CoV polyprotein genes called *ORF1a* and *ORF1b* which together encode at least 16 smaller non-structural proteins (nsp)s when processed by a viral protease. Others include genes for *S* (spike), the two viroporins *ORF3a* and *E* (envelope), *M* (membrane), and *N* (nucleocapsid). Our goal was to align the AA sequences of these proteins to identify covariant pairs and residues within and between core non-structural and structural viral proteins. The alignment resulted in a 9458 AA consensus sequence with only 1.5% of sites being gaps with low coverage and 2% with residue conservation of less than 80%.

When the AA sequences for each protein are aligned and compared, the degree of conservation varies between each protein and also within individual and discrete protein domains. Genes that encode nonstructural proteins (nsp)s with roles in RNA metabolism and genome replication (i.e., nsp12-14) are among the most highly conserved in AA identity. Others that encode nsp2, nsp3, and nsp4 are highly variable. The NTD and RBD of spike show the most significant variability in both residue identity and length. In contrast, the sequence of much of the CTD of the S1 and the entire S2 subunit of spike is highly conserved. The channel-forming envelope viroporin protein is one of the most highly conserved proteins in contrast to the viroporin ORF3a which exhibits high variability in its NTD and other CTD subdomains. We

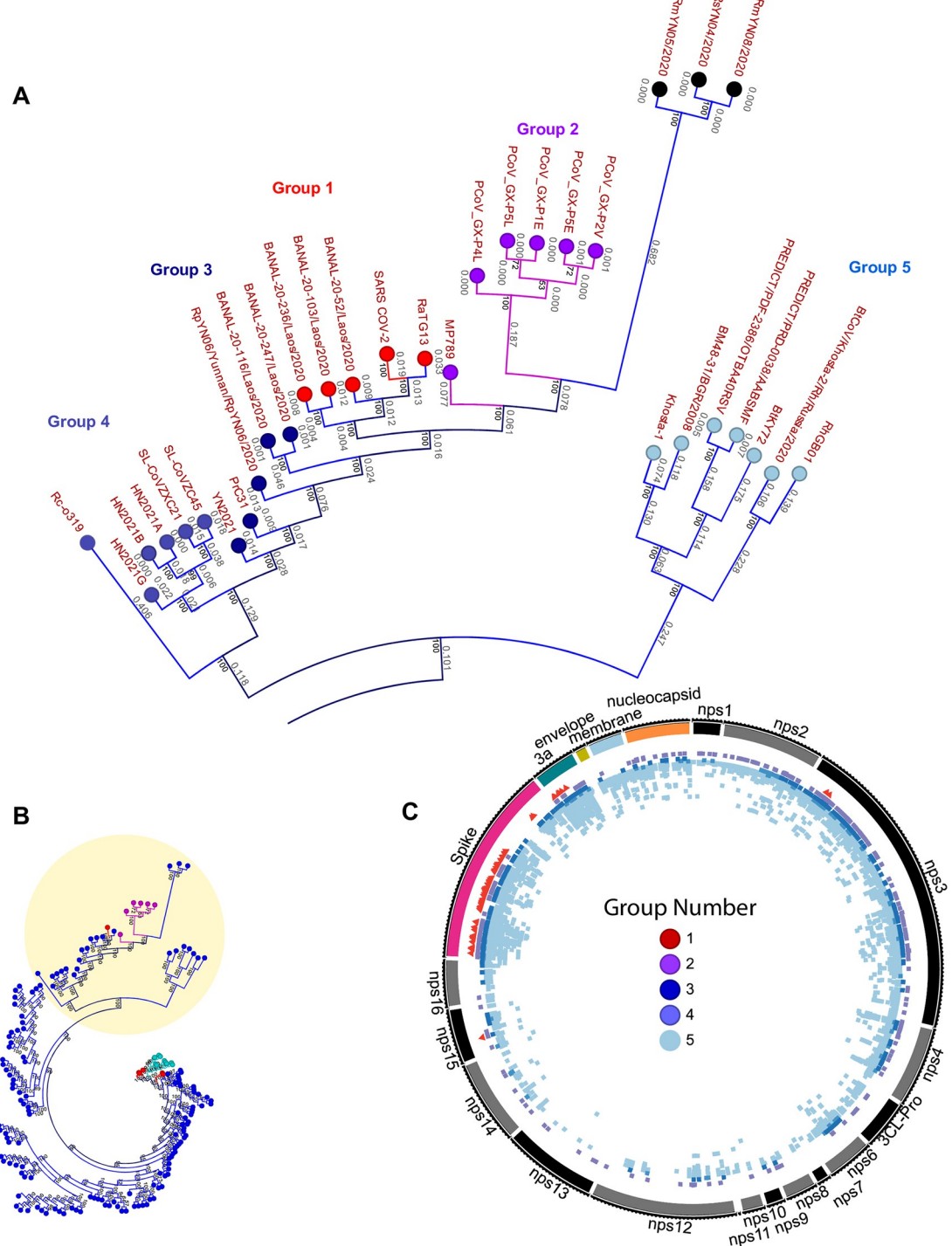

**Fig 2. The distribution of covariant residues in CoVs most closely related to SARS-CoV-2.** (A) the identification and CoVs in Groups 1–5 in a subset of the maximum likelihood tree showing branch length and bootstrap values. (B) Position of tree subset in entire maximum likelihood tree (from Fig 1B). (C) Distribution of covariant residues in core genes for Groups 1–5 based on clusters specific to each group. Each group is indicated by color and the position of every residue is colored.

propose that this is evidence of evolutionary pressure on residues in CoV proteins and that certain protein subdomains may be under higher selective pressure than others.

In addition to greater AA sequence variability in some genes, individual residues and continuous sections of AA are either uniquely present or absent in a portion of CoVs. One key aim of our study was to identify and evaluate the covariance of all residues in our selected proteins among these 149 CoVs without bias. We predicted that his analysis might reveal the existence of critical amino acid residue conservation or changes that would possibly correlate with changes in the virus-host range or biological properties. In this regard, SARS-CoV-2 is recognized to possess unique sequences not present in other closely related CoVs including those that define and enhance a furin cleavage site of the spike protein [25, 33]. After binding to the ACE-2 receptor, cleavage of S by furin and/or other proteases is critical to conformational changes that allow viral envelope-host membrane fusion and subsequent viral RNA entry into the host cytosol [25]. Other changes in S are less understood. For example, certain residues in the NTD of SARS-CoV-2 spike are present in other unrelated CoVs but are notably absent in the corresponding regions of SARS-CoV [55]. Conversely, there are residues in many CoVs with no positional equivalent in SARS-CoV-2. To address gaps in the alignment, we have indicated such residues with a "Z" designation to accommodate possible covariance between residues and such deletion occurrences.

## Extracted networks of covariant residues are informative about evolutionary relationships in CoVs

Covariance is a quantitative measurement of how often the identity of one AA is correlated to the identity of another AA or AAs in either the same protein or in a completely different protein [56–58]. Because covariant AA residues change in concert with each other, they can define critical AA-AA residue interactions within a protein or in homologous or heterologous protein-protein pairs or instead indicate phylogenetic relatedness based on their co-existence [43, 44]. These putative residue interactions may provide new insights into the evolution and relatedness of the CoV family of viruses [59]. To survey the frequency of covariance among a reference collection CoVs, we identified correlative pairs and also assembled groups of three or more (here designated as 'clusters') of covarying amino acid residues using a correlating tandem model [60]. These clusters are not typically generated using other typical pairwise algorithms tailored for determining protein structure or docking interfaces. We chose the FastCoV approach for its distinct quality in identifying larger networks of putative compensatory mutations generated by selection and adaptation which seems well-suited for studying the emergence of viruses similar to SARS-CoV and SAR-CoV-2 [60]. This differs from DCA-based and other covariance approaches used to predict co-evolving residue interactions that may use corrected and weighted correlative data that can also be coupled with other various predictive secondary structure motifs to assist in protein structure and interaction predictions [61]. Our approach simply provides a raw covariance purity and percentage score for pairs and larger networks of covariant residues with no goal for structure-based predictions. We selected a purity threshold (0.7) based on our small sampling size and extracted 973,649 unique pairs and 741 clusters. In this preliminary analysis, we identified a collection of gaps and also unique sequences selectively conserved in some CoVs and we concluded that such deletions or insertions, like residues, may also covary with AA sequences in proteins. Deletions were temporarily substituted as rare alternate amino acids in the alignment and covariance was analyzed to reveal putative covariance between all residues and also deletions. This expanded the total number of unique correlating residue pairs (1,089,836) and clusters (769) (S2 Table). We identified many deletions that correlated with AA residues and also other deletions.

All CoVs genomes, clusters, and residues were graphed using a force mapping algorithm (S1 File, shown in Fig 1A). This interactive graph facilitates the extraction of clusters and respective residues and deletions uniquely present to different groups and subsets of CoVs. Remarkably, the spatial organization of graphed CoVs is highly consistent with our phylogenic estimate based on nucleotide alignment (Fig 1B). CoVs most closely related to SARS-CoV, SARS-CoV-2, and HKU-3 are spatially positioned close to one another in each group solely based on shared covariant residues. Other CoVs that vary regarding relatedness are distributed between these three indicated groups. Because some covariant pairs and clusters are entirely inclusive to single groupings or instead shared between certain CoVs, we conclude that covariant residues can be enriched through a common evolutionary history such as ancestry or can be selected by adaption to a specific host(s).

To provide information about the phylogenic distribution of any cluster that may be due to ancestry, an average taxonomic distribution score (ATDS) was calculated for each cluster based on the number of CoVs present in a given cluster and their average distribution based on branch lengths estimated in the ML tree (S3 Table). Though this score is relative and also determined by the relatedness of all CoVs in the phylogenic reconstructions, clusters and their respective alleles with a larger ATDS are more broadly represented in the evolutionary record within the scope of these 149 β-coronaviruses analyzed. A small ATDS value indicates these covariant residues in a given cluster are restricted to CoVs that are very similar or almost identical. We predict this class of clusters may be biologically informative about covariant residues specifically enriched in SARS-CoV and SARS-CoV-2 and their respective relatives. Conversely, clusters with large ATDS values are those clusters with residues that are present in more evolutionary disjunctively distributed single or groups of CoVs. These may be the result of divergent or independent selective events or are instead conserved covariant residues that have persisted during the evolution of CoVs and are possibly ancestral or even essential to the lineage of these viruses.

Of the 1,089,836 unique pairs with varying degrees of residue identity at every two positions, we identified 522,336 correlative AA residue pair positions and also calculated the number of unique amino acid identities that can exist for each position in the pair (tabulated in S2 Table). This degree of residue representation of each pair varied between a minimum of two (481,024) and a maximum of seven (2). Only ~8% (41,312) of all pairs are represented by three or more unique residue identities and this representation drops significantly stepwise for each unique identity between three and seven. We hypothesized that the increased number of independent residue pairs represented at any two correlative positions in any of these genes in the evolutionary record increases the probability that there is a true interacting relationship between such residues.

The position of every covariant residue in the alignment was mapped to the respective residue position in SARS-CoV-2. For an overwhelming majority of residues that show high conservation and are present among the 149 CoVs, this translational numbering assignment based on the sequence alignments is straightforward. For residues in less conserved regions such as those that exist as gaps or insertions in some CoVs including SARS-CoV-2, this created residues positions that are represented by gaps for sites missing in SARS-CoV-2 and duplicate numbering. For example, several CoVs possess between two and eight additional residues between the aligned numbered AA positions 7 and 8 of SARS-CoV-2 spike. If any of these residues are covariant with other residues or gaps, the use of SARS-CoV-2 residue numbering necessitates either no assignment or the number of the residue that flanks the missing residues in SARS-CoV-2 to preserve the information position of gapped-residue covariance. We chose the latter to indicate the position, thus some residues may appear to be duplicated or even

covariant with themselves when SARS-CoV-2 numbering is employed. We have provided tables that indicate these positions to identify such occurrences (S2 Table).

The presence of more variable covariant residues in other proteins varies significantly. For pairs with at least five independent identities (319), nearly half (154) of these are located in the spike protein. Various structures of spike trimer are elucidated and well-studied due to their roles in receptor recognition, cell entry, and interactions with monoclonal antibodies [23, 62–65]. Using available high-resolution PDB structures, we screened for predicted interacting residues and then referenced our identified covariant pairs to establish a correlation between the number of unique identities in pairs. As observed in the alignment, spike sequences and high order structures vary between CoVs, and we adjusted scoring both for directly interacting residues and those directly adjacent by one residue position (S5 Table). Residue pairs with increased representation are more likely to interact or be in close proximity to one another in the spike trimer protein [~24%] than those represented by only two identities [~5%]. This provided confidence that residues with increased representation are more likely to have direct interactions with their identified cognate pairs.

## AA covariance in the CoVs closely related to SARS-CoV-2 is enriched in spike

We examined the identity and distribution of covariant residues within the lineages of CoVs most closely related to SARS-CoV-2 identified in this work designated as Groups 1–5 (Fig 2A). We reasoned that the selective pressure imposed on the AA identity of truly covariant residues should be different than for all other noncovariant residues. Thus the collection of putative covariant AAs in SARS-CoV-2 and other CoVs that share a common ancestor provides a new perspective on the evolutionary relationships between these viruses.

Group 5 exhibits the most numerous and distributed covariant residues identified in distinct clusters (Fig 2C). This is not surprising based on the apparent evolutionary divergence within these CoVs when compared to Groups 1–4 (Fig 2B). Both Group 2, which is entirely represented by Pangolin CoVs, and Group 3, all bat CoVs, similarly exhibit a greater number of covariant residues when compared to Group 1, also likely due to differences in the overall relatedness of CoVs. The CoVs represented in Group 1 exhibit high conservation and some members are nearly clonal. For these, nearly all covariant residues in this group are restricted to spike and ORF3a, except for two residues in Nsp3 (149 and 175) and a single residue in Nsp15 (115).

Because CoVs in Groups 1–3 are most closely related to SARS-CoV-2 based on their nucleotide identity, we focused on these and examined covariant residues in spike (Fig 3A–3C). Residues in spike are recognized to be among the most relevant in the emergence and persistence of SARS-CoV-2 and also implicated in host adaption in both SARS-CoV and SARS-CoV-2 [31, 66]. We vetted residues in Groups 1–3 because we hypothesized these covariant alleles might be important for selection, adaptation, and viral fitness for the most closely SARS-CoV-2-related viruses in human, pangolin, and bat hosts. Furthermore, we identified covariant residues co-present in either two or all three of these groups. By definition, the AA identity of individual covariant residues is not highly conserved in CoVs, but instead, their conserved identity varies with other residues. Thus any covariant pair or cluster of residues may indicate a direct or indirect conserved interaction between AAs important during the adaption of a CoV. We find a common subset of conserved covariant residues between both bat and pangolin CoVs with those closely related to SARS-CoV-2 in Group 1. These may indicate specific interactions between residues and residue identities especially relevant to the biology of SARS-CoV-2 and related CoVs.

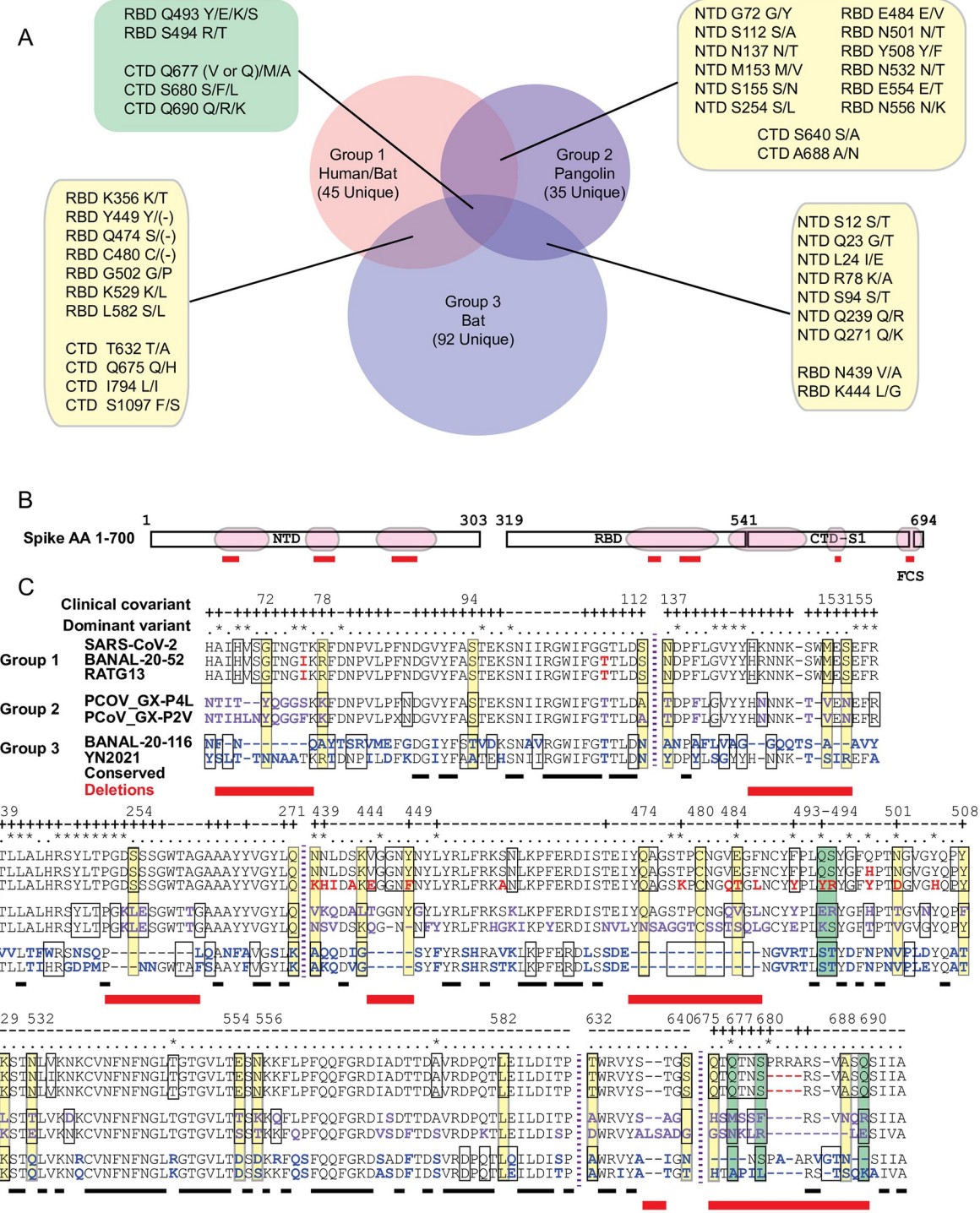

**Fig 3. The distribution of spike covariant residues found in CoVs Groups 1–3.** (A) The number of unique and shared covariant residues between Groups 1, 2, and 3. (B) Overview of covariant residue-enriched domains including NTD, RBD, and FCS that are shown in detail in (C). (C) Aligned AA sequences of covariant residue-enriched regions of spike with representatives from Group 1, Group 2, and Group 3. The residues identical to SARS-CoV-2 are colored black. Residues that differ from SARS-CoV-2 are colored within each of the three groups (red, purple, and blue). Boxed residues indicate those covariant within and between groups. Covariant sequences that are common to two groups are boxed in yellow and those to all three in green and listed in (A). Residues identified as clinical covariants are indicated (+). Residues present in dominant circulating variants are indicated (*). Conserved residues are underlined and residues deleted when these groups are compared marked with a red box underneath the alignment.

A majority of the spike-specific covariant residues common to clusters in Groups 1–3 are located within discrete domains primarily in the S1 domain (Fig 3B and 3C). These regions are in the NTD [AA 67–112, 137–155, and 239–271], RBD [AA 439–508 & 529–589], and CTD of the S1 subunit [AA 632–640], and also at the FCS within the S1/S2 subunit boundary [AA 675–690]. The NTD, RBD, and FCS are also notably enriched in both mutations and deletions identified in dominant variants of SARS-CoV-2. For example, the deletions and flanking mutations at positions 69–70, 142–145, 156–157, and 241–253 found in SARS-CoV-2 dominant variants align well with enriched covariant residues, including deletions, in these three groups. In regions of very sparse covariance such as AA 529–590, two variant mutations 547 (in Omicron) and 570 (in Alpha) also align with covariant residues in Group 1. Conversely, other mutations in current SARS-CoV-2 variants do not align with these enriched covariant residues. True covariant residues require additional compensatory changes at other residue positions and we expect a portion of these residues to be less mutable than other noncovariant residues with low conservation.

## Clinical and pan covariant residues are similar in representation

The resolution and extent of our pan-CoV covariance analysis are in part defined by the number of distinct genomes and also their relatedness. Roles for all identified covariant residues in all proteins cannot be readily ascertained, but this generated data may be further validated as more SARS-CoV-2 protein structures become available. Because of the vast scope and magnitude of SARS-CoV-2 infections during the ongoing pandemic and the availability of whole genomes sequenced, we supposed covariant residues may also be apparent in the millions of sampled clinical strains over 18 months. Genes enriched residues with covariant relationships that appear to be co-present in both the pan and clinical covariant analysis are more likely to be of special interest to SARS-CoV-2 biology. We extracted and stringently selected whole-genome sequences that were nearly or entirely complete to avoid artifacts that may bias our analysis and then compared the positions of covariant residues of 252,102 randomly selected sequences deposited between December 2019 and August 2021 (S4 Table).

Due to the near clonality of SARS-CoV-2 sequences, we were unsurprised to find only 1.2% of the total covariant residues when compared to those identified in the pan-CoV analysis. 13,041 uniquely represented pairs of AA residues can be reduced to 6,137 correlative pairs for all proteins. As observed in the pan-analysis, the distribution is exceedingly skewed toward several genes encoding proteins including spike. When the distribution of every single residue identified in both the pan and clinical analysis is compared gene-by-gene, regardless of the positions of the correlative partner, we discovered the contributed representation for each encoded protein follows a similar trend (Fig 4A and 4B). Genes encoding nsp5 3CL-pro, nsp7-nsp16, Envelope, and Membrane proteins are sparse in coverage of co-identified single residues. In contrast, covariant residues are most abundant in frequency in genes encoding nsp1-3, spike, and ORF3a. Remarkably, in either category, there are proteins and specific regions of proteins similarly enriched in the distribution of covariant residues for both analyses. When the co-occurrence for each residue is quantified for the entire protein, the observed overlap between clinical and pan covariance is found to be statistically significant for nsp2-4, nsp13, nsp16, spike, and nucleocapsid. For proteins with overlap measured to be above the threshold of significance, such as nsp1, envelope, and ORF3a, this graphing allows us to observe both similar patterns and frequencies of covariant residues across the protein. Conversely, for nsp6, nsp10, and membrane proteins we see no significant similarities in residue distribution or patterns.

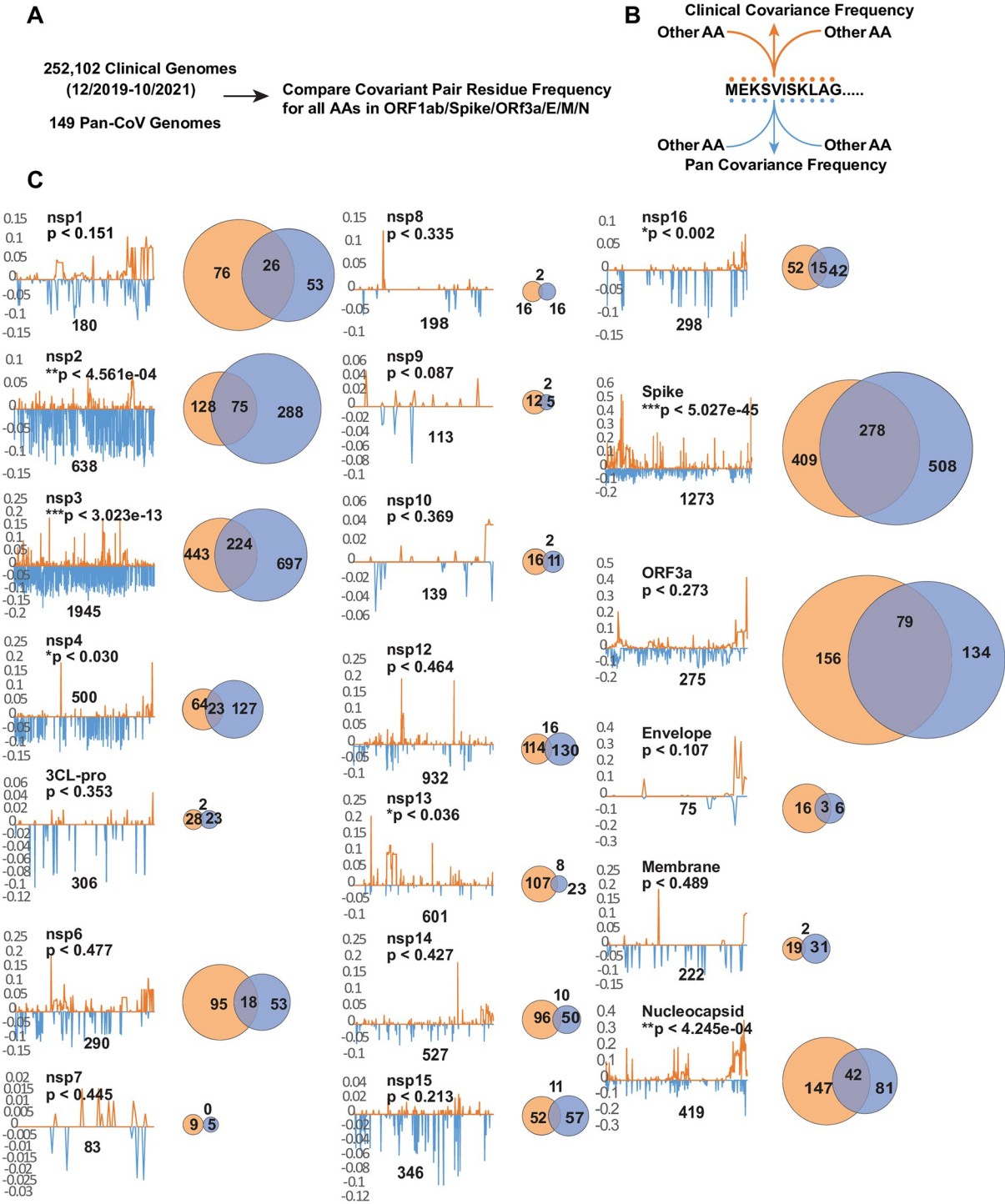

**Fig 4. Comparing individual covariant residues in clinical and pan analyses in conserved genes.** (A and B) Outline of comparative analysis. (C) Trace showing the position and frequency of both Clinical (orange) and Pan (blue) identified covariant residues in each gene. The length of the SARS-CoV-2 gene is labeled and the number of covariant residues in each is indicated with the number overlapping in a Venn diagram. The size of each Venn circle is proportional to the percentage of residues identified in each gene., The significance and P-value calculated for each gene based on the length of the gene, number of covariant residues in each, and overlap is shown.

## Mapping of identified conserved pairs in both analyses

We accounted for the distribution of intra- and inter-protein residue pairs identified in both analyses (Fig 5A). We reasoned these pairs are more informative about conserved residue interactions than the distribution of single residues mapped per protein. As with single residues shown in (Fig 4), the distribution of linked residue pairs is not uniform and the density varies significantly by protein and within protein regions. The majority of linked pairs are mapped to genes encoding nsp1, nsp2, nsp3, spike, ORF3a, and nucleocapsid. For genes encoding nsp4 and nsp16, the occurrence is sparse or absent in residue pair representation. Of 538 total pairs (S6 Table), 90% of residues (485) are represented by residues within the same protein (Fig 5B), are frequently proximal or adjacent to each other by position, and are not random in distribution. We expect this bias as most interacting residues should be present within the same protein. The remaining 10% (53) intra-protein pairs are similarly clustered and nonrandom in their positional enrichment (Fig 5C). Intra-protein residues in nsp3, spike, and nucleocapsid are linked with the most diverse partner proteins (Fig 5F). In contrast, nsp12, nsp13, nsp15, nsp16, and envelope possess only one or two intraprotein pairs.

## Evidence for interactions within and between spike and ORF3a linked to viral emergence and adaption

The enrichment of residue pairs in the subdomains of spike and ORF3a was of special interest (Fig 5D). First, the distribution and abundance of these residues are similar to other covariant residues we identified within the CoVs most related to SARS-CoV-2 (Fig 5C) Furthermore, of these 224 residues, over a third (88) are positioned within spike and Orf3a. Of the 88, 40 are covariant pairs, with 31 identified in dominant variants (S6 Table). When 31 residue pairs are mapped to spike and ORF3a, most links are enriched within the NTD and RBD of spike with two notable links between the spike NTD and AA 26 in the NTD of ORF3a (Fig 5E).

We find evidence that subsets of these 31 residue pairs likely interact directly or are positioned proximal to one other within particular regions of the spike protein (Fig 6A–6E). In a solved quaternary structure of the spike trimer PDB (7JJI.PDB), NTD residue 20 is adjacent to its covariant pair residue 138 in spike Cryo-EM reconstruction (Fig 6D). Moreover, residues 17 and 21 interact directly with 138 Residues between 138 through 157 include identified covariant residues in this work that are notably deletions and/or mutated in dominant variants. Similarly, covariant residues 241 through 252 are also frequently deleted and/or mutated and these directly interact with 138–157. Residues 248–250 identified in our work are covariant with residue 75. With residue 75, deletions and mutations between residues 65 and 82 are also among the most abundant identified in dominant circulating variants. Residues 212 and 215 reside in yet another covariant hotspot and have covariant pairings with residues 142 and 241/242, respectively. Curiously, based on structure, AA residues 212–215 have no apparent direct interactions with 138–157, 241–252, or 67–75. All of these mutation and deletion hotspots in spike NTD have generated much interest regarding their roles as superantigens and the escape from neutralizing antibodies [more below]. The stand-alone identification of these in both pan and clinical covariance analyses and co-occurrence in dominant circulating variants indicate these are often under significant selective pressure to either mutate or become absent by consequence of in-frame deletion, often in the context of distal residues.

Spike protein remarkably accounts for 58% of residues in our identified 538 coincident pairs and also is recognized to possess the majority of mutations in dominant variants. We examined all 538 residue pairs (393 unique residues) and then cross-listed the occurrence of each residue in dominant variants to identify those in all proteins. 119 (30%) of 393 total represented residues are found in 15 dominant variants including the two current Omicron

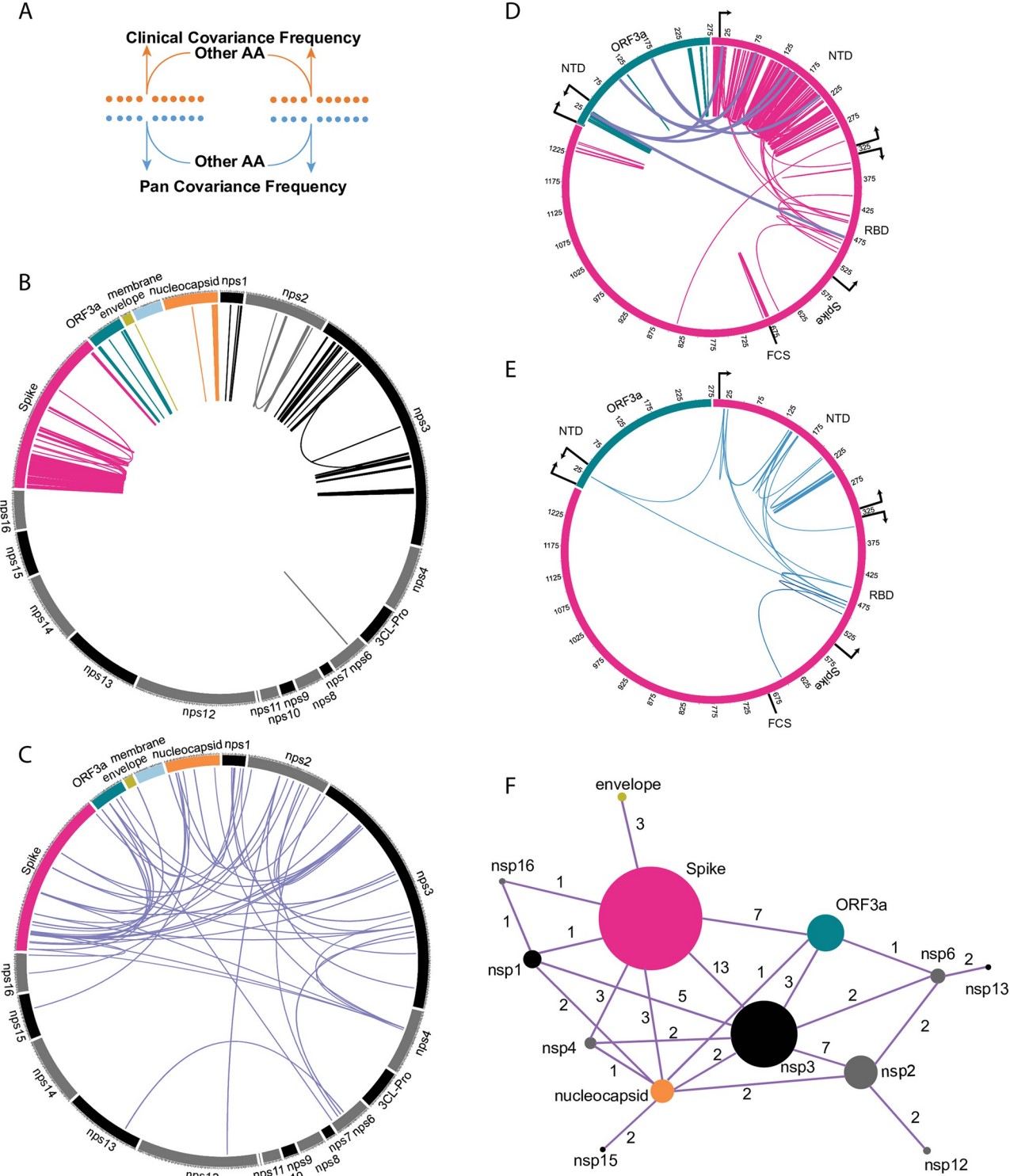

**Fig 5. Mapping covariant pairs common to both clinical and pan analyses.** (A) Diagram showing the concept of conserved residue pair. (B) Distribution of intra-gene covariant pairs. Links are colored by the gene. (C) Distribution of inter-gene covariant pairs. (D) Distribution of Spike and ORF3a inter-and intra-gene covariant pairs. Domains and boundaries of ORF3a (NTD) and Spike (NTD, RBD, and FCS) are shown. (E) Distribution of Spike and ORF3a covariant pairs from (D) that are also present in dominant circulating variants. (F) Network graph showing the number of covariant residues represented in each gene (size of circle) and the occurrence with the number of inter-gene covariant residues between each gene (value indicated for each linkage).

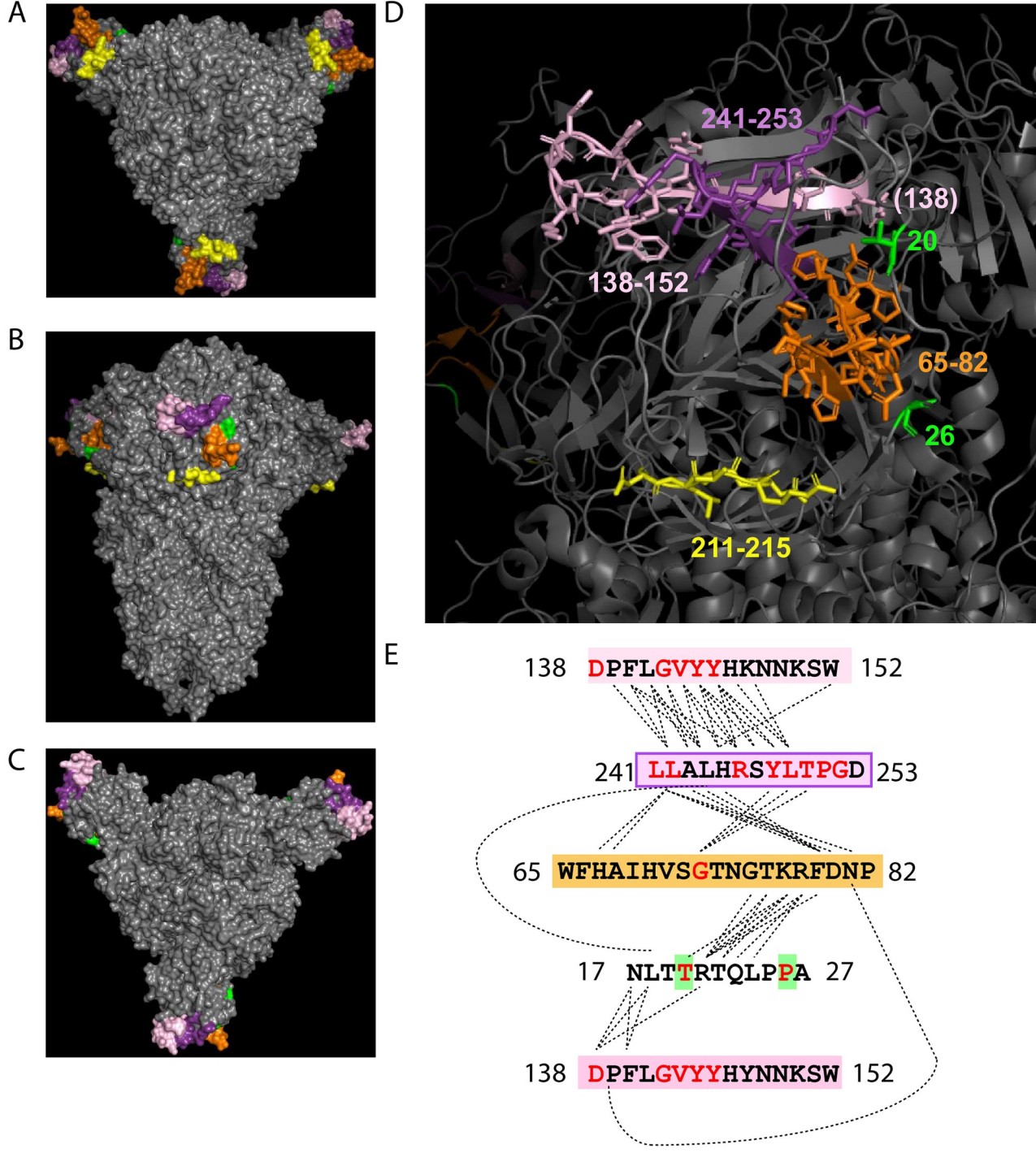

**Fig 6. Mapping of Spike NTD covariant residues and deletion identified in clinical and pan analyses and also found in dominant circulating SARS-CoV-2 variants.** (A-C) The top, side, and bottom PDB structure (7JJI.PDB) of the Spike homotrimer shows the position of these residues. Residues are colored by position in the NTD. (D) Labeled regions and residue numbers of regions are indicated. Amino acid structures are shown for highlighted residues. (E) The sequences of highlighted residues are shown. Residues predicted to directly interact at the molecular level in the PBD are linked by dotted lines. Covariant residues found in both Pan and Clinical covariant analyses and dominant SARS-CoV-2 variants are colored red.

sublineages. When both residue pairs are present in a dominant variant, we find that 37 of these are remarkably present together in the same variant lineage (Fig 7, S6 Table). This observation is suggestive of covariance pressure operative in the emergence of variants during the ongoing pandemic.

## Covariance, immunity evasion, and hACE-2

Numerous studies have characterized SARS-CoV-2 spike mutants that escape antibodies in the context of structure and function (surveyed and reviewed by [67]). Other residues, several that are functional in immune evasion, interact directly with the hACE-2 receptor [23]. As the RBD is especially enriched in such residues, there are presumed constraints imposed on residue mutability here in the context of spike function, hACE-2 receptor interactions, and escape from antibody neutralization. It is not surprising that some enriched mutations in dominant variants enhance cell-receptor interactions, improve spike function, and reduce neutralization by the host immunity, however, these probably operate without imposing significant secondary deleterious effects on viral fitness or have acquired mutations that reduce such impacts. We compared recent Omicron variant mutations in the spike that are demonstrated to circumvent immunity with our pan-analysis to evaluate their frequency of occurrence with other co-present mutations. We reason that only one of the residues in a covariant pair may increase escape from immunity or increase beneficial interactions such as cell receptor recognition and binding. Covariant residues with no known function in these processes may instead compensate in structure or by other yet undiscovered means and therefore facilitate a gain of function (Fig 8B). In most covariant pairs or groups of residues, the identity of which mutation arose first cannot be easily ascertained by the time a dominant variant has emerged and is detected by genomic surveillance. However, within emerged variants that continue to adapt, the stepwise acquisition of additional mutations can yet be determined in sublineages. In Omicron 22A (BA.4), 22B (BA.5), and 22C (BA2.12.1) variants, we find a total of 13 ancillary mutations when compared to the original 21L (BA.2) (Fig 8A). New alleles in spike L452, a residue well- recognized for immunity escape [68], are detected in 22AB (L225R) and 22C (L252Q). Our pan-analysis reveals covariance between 452 and 6 other spike residue mutations (T19, A27, S477, T478, E484, and Q493) co-present in these subvariants with contrasting roles in immunity escape. We conclude that any combination of these 6 residues might have contributed to the selection of L452R/Q and recognize that L452R has always emerged with mutations at 478, 484, or both sites in the Delta, Kappa, and Omicron variants. Like L452 mutations, the deletion of residues at 69–70, which is instead not implicated in immune evasion, has independently emerged in multiple variants and always copresent with mutations or deletions between spike residues 142–144; these are demonstrated to epitopes for neutralizing antibodies [55]. We note that residues 69 and 142 are covariant in our pan-analysis and that the G142D mutation in Omicron 21L preempted the 69/70 deletion in 22A and 22B. Thus a mutation first selected within an epitope might have been improved by a second mutation with no yet known role in immune evasion. Furthermore, the V213G allele, a residue identified covariant with both 69 and 142, is also present in 21L and could conceivably play an additional compensatory or selective role. Curiously, one mutation in Omicron 21L is a revertant in its sublineage; Q493R reverted in 22A and 22B, and Q493 is covariant with L452. Thus, the acquisition of L452R and increase in fitness may have selected the revertant R493Q covariant residue, one operative in both immune escape and hACE-2 interactions. Omicron 22C differs in this respect, the L452Q mutation is maintained as copresent with Q493R. No other dominant emerged variant is yet identified as possessing both L452R and Q493R alleles.

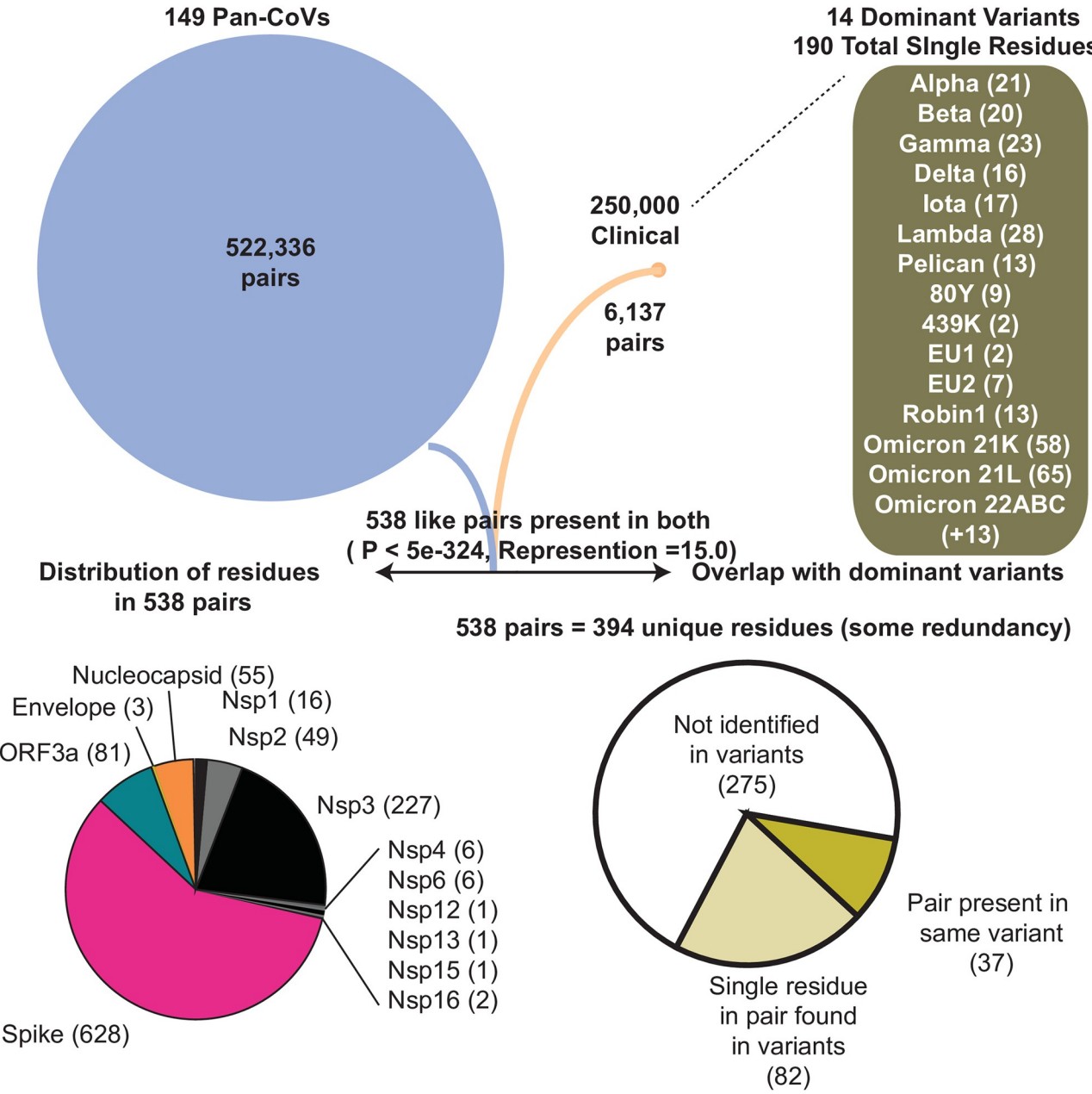

**Fig 7. Flow chart showing the accounting of identified pan and clinical covariant residues and overlap.** 14 dominant variants used in comparative analyses and respective residues found in each are shown in parenthesis. The WHO label for each variant is used for reference. The gene by gene distribution of the 538 co-present pairs in genes is graphed. The abundance and overlap of pairs and single residues identified in the 14 dominant lineages are graphed. Hypergeometric probability of common residues is shown and the representation value indicates the overlap value divided by the number of expected pairs to overlap if all covariant pairs were equally probable.

## Discussion

For both nucleotide and amino acid identity-based approaches, the sequence conservation in either complete or partial regions of CoV genomes continues to be applied to understand the relatedness between β-coronaviruses and SARS-CoV-2. This extends to the emergence of SARS-CoV-2 as a human pathogen responsible for a global pandemic and its continued

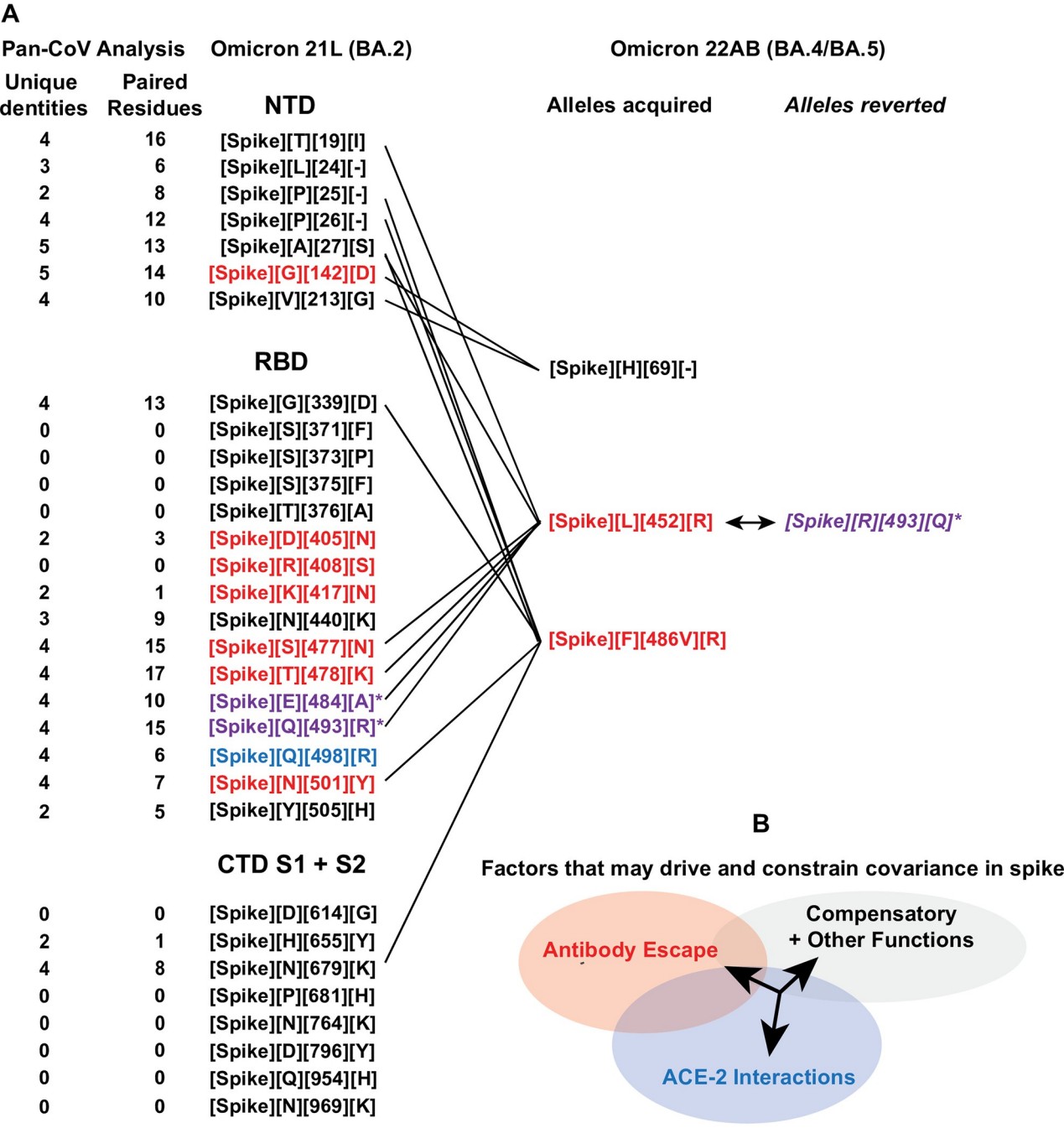

**Fig 8. Comparison of spike protein mutations in Omicron 21L and its two emerged sublineages 22A and 22B.** (A) Spike mutations are listed as separated by protein subdomain in Omicron 21L and 22A/B. Covariant residues and their respective maximum unique identities and the total number of residues paired within the pan covariance analysis are indicated. Residues that are demonstrated to escape neutralization by antibodies or serum are colored red. Those demonstrated to interact directly with the hACE-2 receptor are colored blue. Those demonstrated to do both are colored purple with an asterisk. New mutations that emerged or instead reverted in 22AB are indicated with covariant linkages to residues in 21L and 22AB. (B) A generalized model showing how mutations and compensatory mutations with varying roles of biological significance may be co-selected based on covariant relationships.

adaptation. In this work, we examined the conservation of correlative covariant pairs and even clusters of amino acids that appear to change in concert with one another across the entire genome. We acknowledge apparent covariant mutations can also be a simple consequence of spontaneously emerged mutations and common mutations at less conserved sites. Conversely, these could be a result of spontaneous mutations that imposes a selection for one or even more compensatory mutations at other sites to maintain or even increase viral fitness. Both instances are certain to be present in this dataset. An applied hypergeometric probability distribution predicts the overlap of 538 covariant pairs between the pan and clinical datasets is exceedingly significant (Fig 7). We acknowledge that the conservation and essentiality of amino acid residues vary significantly in CoVs based on the scope of evolutionary relatedness. This should also influence the probability of true coupling because not all residues are equally conserved or mutable. We propose future efforts should examine such covariance in the context of residue mutability. Recent efforts that applied DCA to predict epistatic interactions in the context of SARS-CoV-2 residue mutability concluded coupling also played a minor role in emerging mutations within variants. The authors surmised that a restricted number of unique genomes and a broad scope of evolutionary divergence among all coronaviruses also limited the analysis performance for epistasis-mutation comparisons in SARS-CoV-2 proteins [69]. We chose to limit our pan-analysis to only lineage B β-coronaviruses for our work based on this same principle.

It is important to acknowledge any single nucleotide change can influence viral fitness regardless of its consequence for residue identity and its role in structure and function including covariance. For example, mutations in nucleocapsid found in multiple variants have been recently demonstrated to alter or induce the expression of subgenomic mRNAs that encode a truncated form of the protein that is an inhibitor of type I interferon production [70]. Furthermore, synonymous changes account for approximately 30% of mutations during the pandemic and a recent analysis using 400,000 samples reveals persistent synonymous changes in codon usage appear to be biased toward usage in the human host [71]. We have ignored the aspect of codon usage, cis-acting RNA sequences, and RNA secondary structure in the scope of our current analysis. We propose that integrating such data with AA residue covariance and with other biologically relevant sequence-based information, including predicted recombination events among SARS-CoV-2 and its most closely related CoVs, will strengthen the prediction of the contextual significance and selection of emerging mutations.

We find evidence of true covariance in this work when we compare the total number of independent changes present in each pair with a known structure. In spike, the probability of either a direct or possible indirect interaction by one flanking residue increased from 5% to 24% when the minimum number of unique AA changes in any given residue pair of two identities were compared to those with five (S5 Table). Furthermore, covariant residues with increased independent residue representation in the pan covariance analysis were also more likely to be identified in the clinical covariance analysis. Notably, when residues identified in both analyses are compared, 90% are found within the same gene and enriched in certain genes with important virus-host interactions such as spike. This observation is consistent with an expected model where intraprotein covariance is predicted to be more abundant than that for interprotein, including proteins that form homomultimers such as spike.

A benefit of comparing residues from both covariant analyses is demonstrated by their co-presence and enrichment in dominant circulating variants. Mutations and deletion-enriched hotspots in the spike NTD described in this work are recently identified and studied as antigenic regions responsible for antibody escape [55, 72–76], but not yet investigated comprehensively in the context of pan-covariance to our knowledge. Notably, many NTD covariant hotspots are also deletions identified in SARS-CoV spike protein when aligned to SARS-CoV-

2 [55]. Though clinical SARS-CoV-2 covariance data should reveal co-present residues common to variants, the independent occurrence of these in the Pan-CoV covariance is intriguing. We note these clinical sequences were collected through August 2021, over 3 months before the first emergence of Omicron, and we identify some Omicron-specific residues and pairs in this work. Moreover, contemporary subvariants of Omicron that emerged in the Spring of 2022 such as 22A (BA.4) and 22B (BA.5) have acquired a handful of additional mutations. The majority of these new alleles are indeed strongly covariant with other co-present mutations in Omicron. Recent structural and biochemical work finds significant synergistic fold changes in hACE-2 binding when residues that directly interact with the receptor such as Q493 are co-present with the accumulation of mutations that alter K417, L452, E484, and N501 [77], prominent key mutations also characterized for immune evasion. We find strong pan-covariance among these residues and are intrigued to find L452R accompanied by an R493Q reversion in most current Omicron sublineages. All three sublineages, but most notably those with L452/R493Q, were recently shown to be significantly more resistant to antibody neutralization by both therapeutic monoclonal antibodies and serum from individuals already vaccinated and boosted with three shots of mRNA vaccines [78]. In both current and future variants, we speculate there may be key covariant alleles that when present act as drivers for assemblages of additional mutations important for virulence and immune evasion. In this model, variant mutations accumulate in part due to residue covariance that is conserved and evident in the evolutionary record of both diverse and SARS-CoV-2-related coronaviruses.

We propose all covariant residues identified in both spike and other conserved CoV proteins in this work serve as a reference for possible mutational composition that might arise in current and future dominant variants. These could inform about altered functional domains, antigenic regions, and important epitopes that are more vulnerable to enriched mutations also in part due to their covariant relationships with other residues, even those located in distinct domains or other viral proteins. For example, we speculate some fraction of the mutations and deletions that permit escape from neutralizing antibodies or increase infectivity also require compensatory mutations to maintain fitness and vice versa. Variants that emerged early during the pandemic lacked the opportunity for such selective purification provided by continued serial passage across the global population and the challenges of host immunity conferred by both prior infection and vaccination. The first mutations that emerge early in future variants may serve as a bellwether for the subsequently selected mutations and eventual strain infectivity and severity. Detected residues in new lineages that are likewise found to be significantly covariant with other residues that have been demonstrated to increase virulence, stability, transmission, or immunity evasion can then inform about such issues regarding vaccine efficacy and even clinical outcomes. Furthermore, these may reveal key residues that contribute to yet undiscovered interactions between all viral proteins of SARS-CoV-2 including spike and ORF3a as discussed in an earlier description of our initial results [59].

Future clinical applications such as the concept of pairing single and covariant mutations in SARS-CoV-2 with host polymorphisms or clinical metadata is appealing because such a model is likely to be developed for personalized medicine if this pathogen remains persistent or established in recurrent and seasonal outbreaks. Studies have examined the severity of certain variants in populations dependent on age, health, pre-existing conditions, and even host genetics (e.g., hACE-2 polymorphisms) [79–81]. We did not have access to such metadata for this work and many of the deposited clinical sequences are only annotated for sex, age, and the global location of the patient. Other informative and complementing clinical studies have followed the accumulated mutations over time in immunocompromised patients chronically infected with distinct variants of SARS-CoV-2. These revealed clusters of mutations that included those in residues recognized to be operative in antibody neutralization and receptor binding [72,

82–84]. Some of these mutations that co-emerged within a single patient are also identified as pan-CoV covariants discussed in our work and also present in emerged dominant variants. Including such rare cases, comparative covariance analysis between pan-β-coronaviruses and SAR-CoV-2 during the ongoing global pandemic can now be even extended to those that are detected within the duration of an infection within a single individual.

## Methods

### Genome and protein sequence acquisition and alignment

Nucleotide and protein sequences for the 169 individual CoVs and 252,102 clinical samples were downloaded from available NCBI and GISAID public databases [38, 85, 86]. All genomes and accession numbers are provided in S1 Table. The GISAID sequences have been provided from various sources and published work and these source data are acknowledged in S4 Table. Nucleotide sequences of 169 CoVs were aligned using the MAFFT iterative consistency-based setting (G-INS-i) and we used nucleotides that spanned the aligned start codon of SARS-CoV-2 *Orf1a* gene through the stop codon of the *N* gene [87]. Protein sequences for NSP1 through NSP16, spike, ORF3a, E, M, and N of 149 CoVs were concatenated and then aligned using the MAFFT iterative consistency-based setting (G-INS-i) [87]. For clinical samples, we initially selected a total of 882,364 sequences isolated, sequenced, and deposited in GISAID between December 2019 and August 21, 2021, based on their near completeness (>95%) of sequence coverage for all proteins used in pan-CoV amino acid covariance. This facilitated the inclusion and identification of sequences with small deletions and rare insertions. Due to computational limits, two independent sets of 126,051 randomly selected sequences of the 882,364 were aligned using MAFFT using respective references of each alignment to maintain identical length (List provided in S4 Table) [87]. We expect some deletions and insertions are due to sequencing and assembly errors but only co-varying deletions should become apparent during covariance analysis. The clinical alignment spans both known and predicted genes between nsp1 and ORF9c, but only covariance between the proteins also studied in pan covariance set are analyzed and compared in this work.

### Phylogeny and ATDS calculation

We inferred phylogeny by reconstructing a maximum likelihood (ML) tree with IQTree after first testing and comparing 286 DNA models by creating initial parsimony trees scored according to Bayesian information criterion (BIC) using IQTree ModelFinder [88]. We then applied the best-fit DNA model which is a general time-reversible model using empirical base frequencies allowing for the FreeRate heterogeneity model across sites GTR+F+R6 (invariable site plus discrete Gamma model) with 1000 replicates using bootstrap resampling analysis [88]. This tree file is available in (S2 File). Bootstrap resampling analysis was completed using 1000 replicates. Bootstrap values and branch lengths are indicated in an unrooted tree shown as a circular phylogram. Branch lengths shorter than 0.0368 are shown as having a length 0.0368.

For all genomes that belong to each cluster, the sum of branch lengths between every possible pair was extracted from the tree file and averaged to calculate the Average Taxonomic Distribution Score (ATDS). This relative score is provided as additional metadata in the Gephi Force mapping file.

### Covariance analysis and force mapping

Pairwise and multiple residue covariance and scores were calculated using FastCov [60]. Alignment files for both pan and clinical CoVs were substituted to provide a "W' in place of

absent/deleted residues. Using the known position of true "W", residues, all "W" deletions were replaced as "Z" to indicate absence following analysis. We set a purity score (0.7) for stringency cutoffs in both the pan-CoV and clinical-CoV sequence alignment. A raw table of predicted covariant pairs is provided (S2 Table). This allowed the binning of clusters and respective strains for Force Mapping in Gephi using the Multigravity ForceAtlas 2 setting and comparison of covariant residues based on clusters and strains [89]. All clusters and residues and their respective occurrence in CoVs for both analyses are tabulated in (S3 Table). Genomes, clusters, and residues were mapped in Gephi using the MultiGravity ForceAtlas 2 algorithm. [89]. This data is provided in (S6 Table) for interactive application using Gephi Software.

## Prediction of proximal and interacting residues and mapping of residues in spike trimer structure

The Arpeggio program was used to calculate inter and intramolecular interactions between residues in the 7JJI.PDB file (S5 Table) [63, 90]. To accommodate minor sequence and structure variability between spike proteins in the pan-CoV analysis, the position of any two residues identified to interact in SARS-CoV-2 was extended by one flanking position both amino and carboxyl to each residue when calculating possible interactions for all 149 CoVs. Residues in spike were mapped onto the PDB structure for spike (7JJI.pdb) using PyMol (v.2.3.4) [62, 91, 92].

## Cross-referencing residues present in dominant variants

Mutations identified in previous and dominant circulating variants of clinical interest were extracted from data compiled by CoVariants.org and enabled by GISAID [37, 38, 93]. The WHO label for each variant is used for reference.

## Statistics

Hypergeometric probability was applied in R using the abundance and distribution of single residues in each analyzed gene in the pan and clinical covariance datasets. Residue identity by position was approximated for the pan covariance and then numbered by position in SARS-CoV-2. For comparative analyses of covariant pairs identified in both analyses across all genes, residue identity by position was approximated for the pan covariance and then numbered by position in SARS-CoV-2. The total number of unique residues identified as covariant in each independent analysis and the total pairs co-present (overlap) was examined as above by applying a hypergeometric probability formula using all possible pair combinations between proteins examined in this work.

## Plotting

Circular graphing of key collections of residues was graphically plotted using Circos [94].

## Ethics approval and consent to participate

This study includes sequence and metadata of 252,102 CoV virus strains from a publically available database (GISAID) and though patient age and sex has been approved to be publically available in this database, only the locations and date of virus isolation are noted in this work. No IRB approval is needed for this data or and all acknowledged sources and authors for every sequence in this source data tabulated from the public GISAID are provided in (S4 Table).

## Supporting information

**S1 Table. Genomes and accession numbers of β-coronaviruses used in both phylogenic and pan covariance analysis.**
(XLSX)

**S2 Table. Tables of covariant pairs identified in both pan and clinical covariance analyses.** Raw tables are provided to show purity and percentage for each pair. For each pair in the pan-covariance, the number and identities of different residues are shown. Residue position by alignment consensus and the estimated corresponding position in SARS-CoV-2 are both tabulated. "Z" Residue indicates deletions.
(XLSX)

**S3 Table. Table of clusters, residues, and respective genome names for pan covariance analysis.** Residue numbering is indicated by SARS-CoV-2. The "-"symbol indicates corresponding residues absent in SARS-CoV-2 based on alignment and "Z" symbol indicates deletion. Calculated ATDS for each cluster is tabulated and provided.
(XLSX)

**S4 Table. List and description of both sets of 126,051 sequences extracted from GISAID used for clinical covariance analysis.** Source data for each sequence is provided as a reference.
(XLSX)

**S5 Table. Calculated molecular interactions in the 7JJI.PDB file using Arpeggio.** Chain, Residue, and Atom interactions are indicated. All direct and indirect positions are indicated. All pan covariant pairs for Spike are listed by the abundance of independently identified amino acids and co-occurrence of indirect and direct interactions predicted by Arpeggio are indicated. Accounting of co-occurrence between pan-covariance independent pair representation and predicted interaction in 7JJI.PDB is tabulated.
(XLSX)

**S6 Table. 538 pairs identified that overlap between the pan and clinical analyses.** The count of each residue in the pan CoV analysis is presented. Presence in the selected list of dominant circulating variants is indicated for each pair. The WHO label for each variant is used for reference.
(XLSX)

**S1 File. Gephi force mapping file of residues, clusters, and CoVs from pan-CoV analysis.**
(ZIP)

**S2 File. Newick file for ML tree generated using 30,533 nucleotide alignment of 169 CoVs between SARS-CoV-2 *Nsp1* start site and *N* stop codon.**
(ZIP)

## Acknowledgments

We are especially appreciative and gratefully acknowledge the authors and originating laboratories and hospitals responsible for obtaining the virus specimens and the laboratories where genetic sequence data were generated and shared via the GISAID Initiative (sources provided in S4 Table).

## Author Contributions

**Conceptualization:** William P. Robins.

**Data curation:** William P. Robins.

**Formal analysis:** William P. Robins.

**Funding acquisition:** John J. Mekalanos.

**Investigation:** William P. Robins.

**Methodology:** William P. Robins.

**Software:** William P. Robins.

**Validation:** William P. Robins.

**Visualization:** William P. Robins.

**Writing – original draft:** William P. Robins, John J. Mekalanos.

**Writing – review & editing:** William P. Robins, John J. Mekalanos.

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
