## [Decision Letter · Decision Letter 0]

28 Apr 2022

PONE-D-22-08203Covariance predicts conserved protein residue interactions important to the emergence and continued evolution of SARS-CoV-2 as a human pathogenPLOS ONE

Dear Dr. Robins,

Thank you for submitting your manuscript to PLOS ONE. After careful consideration, we feel that it has merit but does not fully meet PLOS ONE’s publication criteria as it currently stands. Therefore, we invite you to submit a revised version of the manuscript that addresses the points raised during the review process.

We look forward to receiving your revised manuscript.

Kind regards,

Vladimir Makarenkov

Academic Editor

PLOS ONE

Journal Requirements

3. Please upload a copy of Supporting Information Files which you refer to in your manuscript.

Additional Editor Comments:

This paper is relevant and very well written.

I think the authors could add a short discussion (preferably to the Introduction section) about possible origins of SARS-Cov-2. The references that could be cited are as follows:

Boni, Maciej F., et al. "Evolutionary origins of the SARS-CoV-2 sarbecovirus lineage responsible for the COVID-19 pandemic." Nature microbiology 5.11 (2020): 1408-1417.

Makarenkov, V., Mazoure, B., Rabusseau, G. et al. Horizontal gene transfer and recombination analysis of SARS-CoV-2 genes helps discover its close relatives and shed light on its origin. BMC Ecol Evo 21, 5 (2021).

Domingo JL. What we know and what we need to know about the origin of SARS-CoV-2. Environ Res. 2021;200:111785.

Moreover, the authors can also use a new SimPlot++ tool designed to detect recombination and visualize data using sequence similarity networks:

Samson, S. et al. SimPlot ++: a Python application for representing sequence similarity and detecting recombination, Bioinformatics, 2022; btac287 (https://academic.oup.com/bioinformatics/advance-article/doi/10.1093/bioinformatics/btac287/6572334?guestAccessKey=d079b57c-5b8e-4bf4-a1d6-06274bd89169).

Reviewers' comments:

Reviewer's Responses to Questions

**Comments to the Author**

1. Is the manuscript technically sound, and do the data support the conclusions?

Reviewer #1: Yes

Reviewer #2: Yes

2. Has the statistical analysis been performed appropriately and rigorously? 

Reviewer #1: Yes

Reviewer #2: I Don't Know

3. Have the authors made all data underlying the findings in their manuscript fully available?

Reviewer #1: Yes

Reviewer #2: No

4. Is the manuscript presented in an intelligible fashion and written in standard English?

Reviewer #1: Yes

Reviewer #2: Yes

5. Review Comments to the Author

Reviewer #1: This is a nicely written manuscript that exploits the evolutionary history of CoV viruses to identify covariant amino acids (pairs and clusters) as a means to identify interactions within and between viral proteins. This is a fascinating and potentially powerful approach.

One question the authors may want to address in the discussion is the extent to which their analysis could be paired with data on host polymorphisms - in ACE2, IR genes, etc - and clinical metadata, to predict outcomes of COVID-19?

Reviewer #2: Overall, this is an excellent study with stunning analyses and beautiful data viz. However, the authors could make their study more impactful by taking time to integrate the latest --albeit that it's a fast moving field-- findings concerning the biological basis for the instances of co-variance they observe in lineage B beta coronaviruses.

My comments and suggestions for improvement are largely minor:

1. RATG13 is usually rendered "RaTG13"..

2. the authors did spend some time covering the biological implications of certain of the instances of amino acid changes where covariance was observed. Additional work to integrate the narration of covariances with selective drivers, such host species or tissue-tropism specific differences in cell biology / entry dynamics, immunology, evasion of adaptive or innate immune responses et cetera would be helpful. For example, a very recent preprint documented effects of certain non-synonymous mutations within the N gene as in fact being driven by selective advantages conferred by their effects on transcription of viral subgenomic RNAs, not the nucleocapsid protein itself: Mears HV.. et al. (Bauer DLV). 2022. Emergence of new subgenomic mRNAs in SARS-CoV-2. bioRxiv. https://doi.org/10.1101/2022.04.20.488895

3. As submitted, the authors' manuscript appears to be in violation of the GISAID terms of use... but this is only due to what appears to be an innocent technical error: Under the methods (second sentence) the authors write that: “the GISAID sequences have been provided from various sources and published work and these source data are acknowledged in Supplemental Table S4" However, Supplemental File S2 and Supplemental File S4 are identical. And the content of Fig S4 matches that of what is described for SI File S2 “Tables of covariant pairs identified in both pan and clinical covariance analyses”. I do not see a file for the “List and description of both sets of 126,051 sequences extracted from GISAID used for clinical covariance analysis. Source data for each sequence is provided as a reference”

6. PLOS authors have the option to publish the peer review history of their article (what does this mean?). If published, this will include your full peer review and any attached files.

Reviewer #1: No

Reviewer #2: **Yes: **Jeremy P. Kamil

---

## [Author Response · Author response to Decision Letter 0]

2 Jun 2022

Dear Dr. Makarenkov,

Please find our revised manuscript, figures, and uploaded supplemental files through the PLOS One portal. We appreciate the editorial and reviewer feedback and this prompted some changes including the presentation of new data based on the Omicron subvariants that have only emerged within a time window that spans our manuscript preparation during the first half of 2022. We are constantly surprised and amazed at how well this pathogen manages to evolve and adapt faster than our comprehensive current literature can document. For example, differences between Omicron 21L (BA.2) and Omicron sublineages 22AB (BA.4/5) that have only emerged since February 2022 are demonstrated to be important in virulence and immunity evasion of both monoclonal antibodies and polyclonals in the serum of vaccinated individuals. We predict key differences in 21L and 22AB may be indeed due to covariance! This is illustrated in Figure 8 and the evidence is discussed in the new results subsection. Please do not hesitate to reach out with any questions and concerns. The responses to comments and suggestions provided by you and both reviewers are provided below.

We also made a very subtle change to the title ("for" replaced by "to" following important) to correct the grammar. The new title is : Covariance predicts conserved protein residue interactions important for the emergence and continued evolution of SARS-CoV-2 as a human pathogen 

Best Regards,

William Robins

In response to the Editor:

We have edited the format of the manuscript, references, and figures to meet PLOS ONE’s style requirements. Our ethics statement is included in the methods. The PACE tool has been used to check and format all figures. Figure legends have been inserted into the main body of the text.

We added a short discussion into the introduction regarding the possible origins of SARS-CoV-2. Numerous papers that vary in the context of their empirical and editorial content address both strong and circumstantial evidence for numerous theories that explain the emergence of the pathogen in late 2019. Our work provides but one additional perspective that might be supportive and consistent with an eventual elucidation of emergence, but we desire to remain neutral regarding any conclusions unless supporting evidence is remarkably significant. We also are concerned the press often misrepresents and misconstrues the significance of scientific findings to present a more sensational narrative to the public. We hope you find the scope and tone of this added discussion accurate and appropriate for the manuscript.

During the analysis and preparation of the manuscript, we often considered the contributing roles of both recombination and covariance. One large section of a genome that recombines with another related CoV in a co-infected host such as a bat or pangolin may require many back and forth compensatory changes both in new and preserved regions to regain fitness. This could help explain some of the founding mutations needed for SARS-CoV-2 to emerge. As shown in Figure 2C, the distribution of covariant residues in 5 groups of strains most closely related to SARS-CoV-2 is in part correlated to the relatedness within each group. This explains the abundant residues in Group 5 as they are more divergent in residue variability. Conversely, the close relatedness within group 1 reveals fewer residues within these that are enriched significantly in spike, a key gene predicted to be prone to recombination events. Remarkably in all groups, there is distinct demarcation at several sites in the genome where covariance increases or is instead sparse or absent. We surmise this analysis reveals evidence of recombinant events that moved a large number of covariant residues at once such as in spike, ORF3a, and specific regions within Orf1ab. We propose this analysis could stand alone as a follow-up informatics paper and will evaluate the SimPlot++ tool that is conveniently recommended.

In the revised manuscript, there is a new subsection added to the end of these results with an accompanying figure (Fig 8) and supporting paragraphs in the discussions based on helpful suggestions from both reviewers (see below). Published work relevant to these topics is cited within the newly added text.

In response to the Reviewer #1:

We appreciate the comments and suggestions. The pairing of covariant mutations in SARS-CoV-2 with host polymorphisms or clinical metadata is an appealing concept because such a model is likely to be someday developed for personalized medicine regarding persistent or recurrent and seasonal epidemics of such viruses as SARS-CoV-2. Moreover, the severity of certain variants in populations with pre-existing conditions such those immune-compromised and or host genetic variations (e.g. hACE-2 polymorphisms) are likely to be examined in both contemporary and future work. We do not have access to such metadata and many of the deposited clinical sequences are annotated only for sex, age, and location. However, how these mutations correlate with any host factors and various aspects of COVID disease is an important consideration. We added a paragraph to the end of the discussion section and also referenced current research especially relevant to these specific topics. A separate informative approach would be to follow the accumulated mutations over time in rare immunocompromised patients chronically infected with distinct variants. These studies were published earlier in the pandemic and revealed several mutations now known to be operative in immune evasion. Some collections of such mutations were also covariant based on our work.

In response to the Reviewer #2:

We appreciate the suggestions and the kind comments regarding our analysis and figures. 

“RATG13” is now corrected to the commonly used format “RaTG13”. 

The roles of covariant residues in the context of selective drivers are likely very relevant and one primary reason many are present in dominant variants. We added a subsection in the results, “Covariance, immunity evasion, and hACE-2” to detect possible processes of enrichment. During our manuscript preparation, Omicron 21L has since evolved into sublineages including 22AB defined mostly by changes in spike. We examined these emerging spike alleles in the context of covariance and find a number of these new mutations are indeed covariant to other mutations in the original variant. Remarkably, we find one reversion in 22AB in a covariant residue (Q493) that is important in ACE-2 and immunity evasion but possibly constrained my other co-present alleles in these variants. This is diagrammed in a newly added Figure 8. Regarding the influence of mutations that alter biological processes that are independent of protein structure, function, and epitopes, we recognize other consequences related to mRNAs and even codon usage should impose additional selective pressures. The identified subgenomic RNA in the N protein is another example that demonstrates mutations are not always important in the context of proteins and this work is now cited in this manuscript. We have plans to overlay such discovered cis-acting genomic elements, mRNA structure, and mutational biases including apparent codon usage with our covariance data.

The supplemental table S4 file that lists and acknowledges GISAID sequences was correctly prepared, but we suspect the link to the correct file was incorrectly completed by us during the initial upload. We will ensure the correct file is either linked or instead uploaded to PLOS One. We are grateful that the reviewer recognized this problem.

---

## [Editor Report · Decision Letter 1]

8 Jun 2022

Covariance predicts conserved protein residue interactions important for the emergence and continued evolution of SARS-CoV-2 as a human pathogen

PONE-D-22-08203R1

Dear Dr. Robins,

We’re pleased to inform you that your manuscript has been judged scientifically suitable for publication and will be formally accepted for publication once it meets all outstanding technical requirements.

Kind regards,

Vladimir Makarenkov

Academic Editor

PLOS ONE

Additional Editor Comments (optional):

The authors have addressed appropriatly all the comments of both reviewers and of the academic editor.
---

## [Editor Report · Acceptance letter]

13 Jun 2022

PONE-D-22-08203R1 

Covariance predicts conserved protein residue interactions important for the emergence and continued evolution of SARS-CoV-2 as a human pathogen  

Dear Dr. Robins:

I'm pleased to inform you that your manuscript has been deemed suitable for publication in PLOS ONE. Congratulations! Your manuscript is now with our production department. 

Kind regards, 

on behalf of

Dr. Vladimir Makarenkov 

Academic Editor

PLOS ONE